# Evaluating *Dendrolimus superans* (Lepidoptera: Lasiocampidae) Occurrence and Density Modeling with Habitat Conditions

**Daxiao Han** [1,2], **Shuo Wang** [1,2], **Jili Zhang** [1,2], **Rong Cui** [1,2] and **Qianxue Wang** [1,2,*]

1   Harbin Research Institute of Forestry Machinery, National Forestry and Grassland Administration, Harbin 150086, China; handaxiao0818@126.com (D.H.); wangshuo1504@163.com (S.W.); xtafktj@126.com (J.Z.); cuirong930218@163.com (R.C.)
2   Research Center of Cold Temperate Forestry, Chinese Academy of Forestry (CAF), Harbin 150086, China
*   Correspondence: wqx890711@163.com

**Abstract:** *Dendrolimus superans*, a prominent forest pest in northeast China, exerts detrimental effects on tree growth and development, disrupts the ecological functioning of forests, and even alters the trajectory of succession. The objective of this study was to investigate the influence of habitat conditions on the occurrence probability and density of overwintering *D. superans*, aiming to provide scientific insights for the effective prevention of and control measures against this pest infestation. The investigation encompassed 142 plots (20 m × 20 m) in various forest types within the primary distribution area of *D. superans* in the Great Xing' an Mountains, focusing on factors such as topography, forest vegetation, and larval density. Binary logistic regression was employed to establish models for predicting the occurrence probability of *D. superans*, while generalized linear models (GLMs) and categorical regression (CATREG) were utilized to develop models for estimating its population size. Subsequently, an evaluation was conducted to assess the performance of these models. The occurrence probability model showed high accuracy (AUC = 0.826) in predicting infestation. The slope aspect and herb cover were the key factors affecting the occurrence of *D. superans*. The occurrence probability was the lowest on shady slopes and the highest on sunny slopes. The occurrence probability of *D. superans* increased with the increase in herb cover. The model of quantification showed that the density of *D. superans* was the least on shady slopes and the highest on sunny slopes. As the slope gradient increased, the density decreased. *D. superans* occurred most frequently on ridges. Similarly, with the increase in canopy cover or the decrease in diameter at breast height (DBH) and stand density, the density of *D. superans* increased. The influence of the topography factors surpassed that of the forest vegetation factors in shaping the population dynamics of *D. superans*, despite both being significant contributors. The study revealed that *D. superans* is prone to occur on sunny slopes, flat slopes, and ridges, which should be the focus of prevention and control in forest management practices, such as replanting, thinning, and regular weeding, to help restrain the growth of the population of this pest.

**Keywords:** *Dendrolimus superans*; population size; logistic regression; generalized linear model; categorical regression; outbreak mechanism



## 1. Introduction

*Dendrolimus* spp. are classified as Lepidoptera and Lasiocampidae and primarily feed on conifers such as *Picea* spp., *Abies* spp., *Pinus* spp., and *Larix* spp. This significant pest poses a serious threat to boreal forests and continues to expand its distribution [1,2]. An extreme outbreak of *D. superans* occurred from 1989 to 1991, resulting in forest damage exceeding one million hectares in the Great Xing' an Mountains [3]. The average insect population density was approximately 283 individuals per tree, with a potential maximum of up to 2000 per tree [4]. Furthermore, the escalating global warming phenomenon is anticipated to increase the risk of *D. superans* outbreaks in the Great Xing' an Mountains

due to amplified aridity and elevated temperatures [5]. Insects play a crucial role in forest ecosystems as integral components and potential sources of internal disturbances. When they exist in a state of dynamic equilibrium, they contribute to the overall stability of the system. However, during outbreaks or disasters, their presence can result in extensive tree mortality, significantly impacting species composition, spatial structure, and ecological functioning within the forest system [6,7].

At the macroscopic level, insect reproduction and distribution are directly or indirectly influenced by climatic factors such as temperature and precipitation [5,8,9]. Within the realm of community ecology, stand characteristics and soil properties exert a pivotal influence on the occurrence of forest pests and diseases [10]. The pest of the Siberian silk moth (*Dendrolimus sibiricus*) is prone to outbreaks on southwest slopes [11]. A study conducted by Cescatti et al. demonstrated a significant correlation between the elevation of spruce *Picea abies* stands and the infestation levels of *Cephalcia* spp. [12]. The occurrence sites of pine caterpillars (*Dendrolimus* spp.) primarily encompass areas characterized by poor soil quality or exposed rock formations. The host plants exhibit limited growth and possess a diminished resistance to pine caterpillar infestations, which results in more severe defoliation when they are subjected to insect attacks [13]. Insects exhibit habitat selection based on their unique ecological preferences, which leads to the regionalization and fragmentation of their outbreaks within the same community. Therefore, it is of utmost importance to establish models that elucidate the factors influencing the occurrence and quantity of insect pests, enabling their scientific management and long-term monitoring.

The resource concentration hypothesis posits that oligophagous insects exhibit a higher propensity to inhabit pure forests characterized by concentrated host distribution [14]. In addition to impacts on biodiversity, phytophagous insects, which are closely associated with plant growth processes, exhibit a preference for depositing eggs on vigorously growing hosts [15]. The survivorship of insects exhibits a decline as one moves from closed canopy equatorial forests to more deforested habitats, as determined through a habitat heterogeneity experiment employing artificial Play-Doh caterpillars for measuring survivorship [16]. Other studies suggest that the occurrence probability of oligophagous insects tends to increase with an increase in host plant density; however, beyond a certain threshold, the growth rate of the occurrence probability diminishes as plant density increases, indicating the presence of a resource dilution effect [17–19]. The destruction of pine needles in a forest significantly impacts the population dynamics of pine caterpillars, leading to a decrease in the larval nutrient index subsequent to feeding on damaged pine needles [20].

Currently, biological control serves as the primary method of pest management for this species; however, its application in forest management remains limited due to the inadequate consideration of *D. superans* ecological habits and habitat requirements. Thus, the objectives of this study were (1) to identify the influence of topography and forest vegetative variables on the occurrence of *D. superans*; (2) to evaluate the relative importance of these variables and determine the key variables affecting the population dynamics of *D. superans*, thereby providing scientific insights for the effective prevention of and control measures against this pest infestation.

## 2. Materials and Methods

### 2.1. Study Area

The study site was located in the forest farm of Chaoyuan Forestry Bureau, Inner Mongolia, China (120°17′~121°40′ E, 47°35′~48°37′ N). It spanned approximately $3.13 \times 10^6$ ha, with a forest coverage rate of 83.7% and an elevation ranging from 750 to 1450 m. The region is situated at the southern foothills of the Great Xing' an Mountains, characterized by a cold temperate continental climate. It experiences cool and humid summers, long periods of winter snow and ice cover, with an average annual temperature of −2.1 °C, an average annual precipitation of 461.9 mm, and a frost-free period lasting for 87 days. There are 214 genera and 390 species of common plants belonging to 60 families present in this area, which exemplify the coniferous forest ecosystem found within the cold

temperate zone. The predominant species include *Larix gmelinii* (Rupr.) Kuzen and *Betula platyphylla* Suk. The associated tree species primarily comprise *Quercus mongolica* Fisch.ex Ledeb., *Betula dahurica* Pall., and *Populus davidiana* Dode, among others. The dominant understory vegetation encompasses *Ledum palustre* L., *Vaccinium vitis-idaea* L., etc.

### 2.2. Plot Design

In 2019, sample plots were selected from forest farms managed by the Chaoyuan Forestry Bureau (Cuiling, Lizishan, Qingling, Sugehe, and Yulin Forest Farm). Based on second-class survey data provided by Chaoyuan Forestry Bureau for each small class forest group, near-mature forests were chosen as the target small class. A total of 142 temporary survey plots with different forest types measuring 20 m × 20 m were randomly established. The investigation encompassed the analysis of slope aspect, gradient, position, and elevation for each sample. The slope aspect was categorized into five types: flat, shady slope, semi-shady slope, semi-sunny slope, and sunny slope. The slope gradient was classified into four categories: <6°, 6~15°, 16~25°, and >25°. The slope position was further divided into valley, lower slope, upper slope, and ridges.

### 2.3. Investigation Method

In late April 2020, prior to any damage inflicted on the trees by larvae of *D. superans*, a five-point sampling method was employed to select two trees of *L. gmelinii* in each plot, situated at the four corners and near the center, and a total of ten larch trees were chosen. The lateral branches below the diameter at breast height (DBH) of the target trees and the shrub branches connected to their trunks were pruned, while the thick bark at DBH of each tree was scraped off. Subsequently, a closed tape ring was affixed using plastic tape at this location, followed by an even application of shellac onto the tape to create a sealed isolation ring with a width measuring 15 cm (Figure 1). After the larvae initiated their ascent on the tree, the larval number on the enclosed circumference was recorded every 3 days, with subsequent removal and elimination of the larvae until no new insects were detected on the tree for a continuous period of 3 days. The final count of the larvae present on each tree was conducted at the conclusion of the observation. Additionally, measurements were taken of the DBH and height of each tree, while forest habitat factors such as stand density, canopy cover, and herb coverage were documented. Stand density was determined by enumerating the trees with a diameter ≥5 cm within the designated plot.

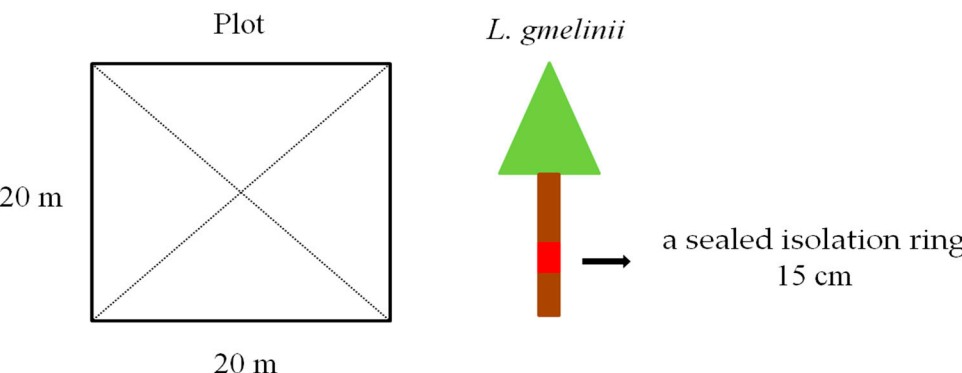

**Figure 1.** Diagram illustrating the survey methodology.

### 2.4. Data Analysis

The number of pine caterpillars in 142 plots was determined; a random selection of 10 trees was made to determine the total count of *D. superans*. Subsequently, the average value was computed as an indicator for the density of *D. superans* on each tree in each plot.

By employing the binary logistic regression model, we considered the presence of *D. superans* in the sample plot as the dependent variable, while topography and forest vegetation were regarded as independent variables. Notably, categorical variables such

as slope aspect, slope gradient, slope position, and stand type were treated as categorical items that required categorization (Table 1). Quantitative processing involved utilizing an exponential function (0, 1), wherein any significant effect ($p < 0.05$) observed for a particular category of qualitative variables warranted its inclusion in the model. By evaluating the discriminative performance of the prediction results, a confusion matrix was generated to assess the model accuracy. Sensitivity and specificity measures were calculated, followed by the construction of a ROC curve to visualize the predictive power. Additionally, the area under the ROC curve (AUC) was computed as an indicator of the model accuracy. The basic formula of a binary logistic regression model is as follows:

$$P_i = F(y) = \frac{e^{a+\sum_{i=1}^{n} \beta_i x_i}}{1 + e^{a+\sum_{i=1}^{n} \beta_i x_i}} \tag{1}$$

After transformation of Equation (1), the linear regression model formula l the function of occurrence probability and the independent variable is as follows:

$$\ln \frac{P_i}{1 - P_i} = \alpha + \sum_{i=1}^{n} \beta_i x_i \tag{2}$$

In Formulas (1) and (2), $P_i$ is the probability of *D. superans* occurrence; $y$ is the dependent variable, indicating whether *D. superans* occurs in this plot, with occurrence = 1, and non-occurrence = 0; $x_i$ is the independent variable, representing the $i$ influencing factor; $\beta_i$ is the regression coefficient of the independent variable.

In 126 sample plots where *D. superans* occurred, we employed generalized linear models (GLMs), considering the density of *D. superans* in the sample plots as the dependent variable, while incorporating the topography and forest vegetation factors as independent variables into the quantitative models. The determination coefficient ($R^2$), root-mean-square error (RMSE), and mean absolute error (MAE) were employed to assess the reliability and accuracy of these models, thereby enhancing the professional and academic expression of the paper. The models were compared using the Akaike's information criterion (AIC), and models with an AIC difference < 4 were judged equally valid [21,22].

The topography and forest vegetation variables that significantly influenced the population of *D. superans* were analyzed using categorical regression (CATREG), aiming to investigate their relative importance in population dynamics. The dependent variable was subjected to a logistic transformation to ensure a normal distribution. The continuous explanatory variables were standardized prior to the statistical analysis. Binary logistic regression model, ROC curve, and CATREG analyses were performed obtained using SPSS Statistics 21.0 (IBM, Inc., Armonk, NY, USA). The GLMs were fitted in the package 'lme4' in R 4.2.3 software (http://www.r-project.org/).

**Table 1.** Topography and forest vegetation factors used as classification criteria.

| Item | Code | Category Hierarchy | | | | |
| --- | --- | --- | --- | --- | --- | --- |
| | | 1 | 2 | 3 | 4 | 5 |
| Slope aspect | $x_1$ | Flat | Shady slope | Semi-shady slope | Semi-sunny slope | Sunny slope |
| Slope gradient | $x_2$ | <6° | 6~15° | 15~25° | >25° | — |
| Slope position | $x_3$ | Valley | Lower slope | Upper slope | Ridge | — |
| Stand types | $x_4$ | I | II | III | IV | — |

Shady slope: north and northeast slopes ranging from 337.5° to 22.5° and from 22.5° to 67.5°; semi-shady slope: east and northwest slopes ranging from 67.5° to 112.5° and from 292.5° to 337.5°; semi-sunny slope: west and southeast slopes ranging from 112.5° to 157.5° and from 247.5° to 292.50°; sunny slope: south and southwest slopes ranging from 157.50° to 247.50°; <6°: flat slope; 6~15°: gentle incline; 16~25°: moderate gradient; >25°: steep inclination. Elevation (m): $x_5$; Tree height (m): $x_6$; DBH (cm): $x_7$; canopy cover: $x_8$; herb coverage: $x_9$; stand density (tree·hm$^{-2}$): $x_{10}$; I: pure larch forest; II: larch–white birch mixed forest with larch as the dominant species; III: white birch–larch mixed forest with white birch as the dominant species: IV: mixed forest with a few larch species. The independent variables in the following models refer to Table 1.

## 3. Results

### 3.1. Model and Evaluation of D. superans Occurrence

The occurrence of *D. superans* was significantly influenced by the slope aspect and herb coverage ($p < 0.05$). The model was derived by introducing the constants and regression coefficients from Table 2 into Formula (2):

$$P(\text{y} = 1|x) = \frac{1}{1 + e^{-(-4.344 - 1.521x_{11} - 3.165x_{12} - 2.733x_{13} - 2.630x_{14} + 0.110x_9)}} \tag{3}$$

**Table 2.** Estimated results of the *D. superans* occurrence model.

| Item | Category | *B* | S.E. | *p* | Exp (*B*) |
|---|---|---|---|---|---|
| Constant | – | −4.344 | 2.184 | 0.047 | 0.013 |
| | Flat | −1.521 | 1.561 | 0.330 | 0.218 |
| | Shady slope | −3.165 | 1.208 | 0.009 | 0.042 |
| Slope aspect ($x_1$) | Semi-shady slope | −2.733 | 1.303 | 0.036 | 0.065 |
| | Semi-sunny slope | −2.630 | 1.223 | 0.032 | 0.072 |
| | Sunny slope | 0 | – | – | – |
| Herb coverage ($x_9$) | – | 0.110 | 0.029 | <0.001 | 1.116 |

Category, refer to Table 1.

The model (3) demonstrated that the occurrence probability of *D. superans* was at its lowest on shady slopes and at its highest on sunny slopes. Furthermore, this probability increased in tandem with herb coverage. The AUC of 0.826, ranging from 0.717 to 0.939, indicated excellent prediction performance and high accuracy of the model (Figure 2).

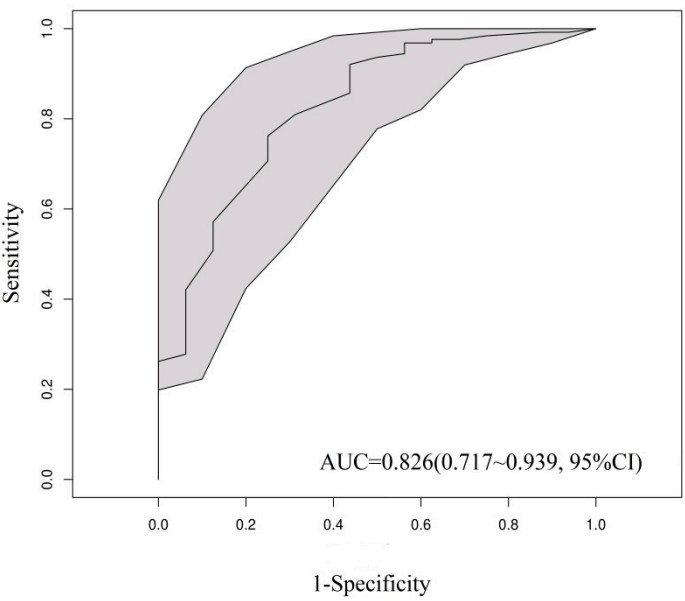

**Figure 2.** ROC curve of the occurrence probability of *D. superans*.

### 3.2. Model and Evaluation of the Dendrolimus superans Population

The density of *D. superans* was found to be influenced by several key variables, including slope aspect, slope gradient, slope position, DBH, canopy cover, and stand density. Our analysis revealed an extremely positive correlation between the density of *D. superans* and the slope aspect ($p < 0.001$), while an extremely negative correlation was observed with slope gradient and stand density ($p < 0.001$). Additionally, we observed a positive correlation with both slope position and canopy cover ($p < 0.05$). Interestingly, there was also a negative correlation between the density of *D. superans* and the DBH ($p < 0.05$) (Figure 3).

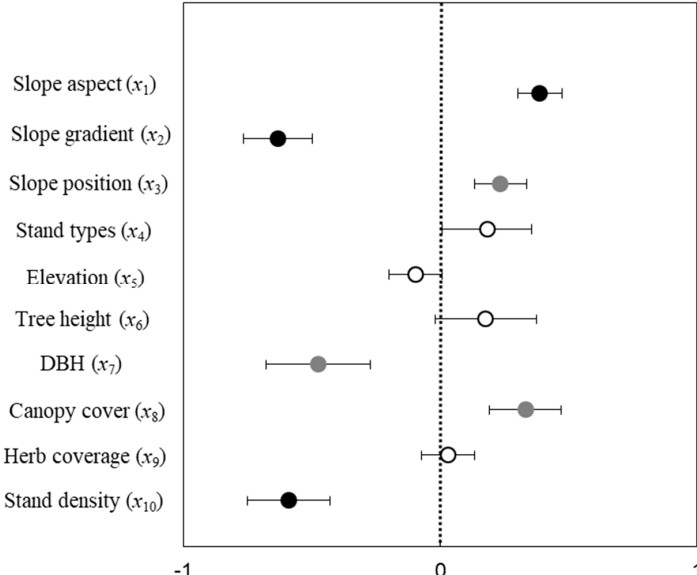

**Figure 3.** Standardized parameter estimates (±SE) of the effects of all topography and vegetation variables on the population of *D. superans*. Black shapes indicate extremely significant effects ($p < 0.001$), grey shapes indicate significant effects ($p < 0.05$), and white shapes indicate no significance.

The six key variables were utilized to re-establish a GLM of the *D. superans* population, revealing significant effects on the population dynamics. Notably, the key variables were consistently observed to have both positive and negative impacts in accordance with the comprehensive variable model (Figure 4).

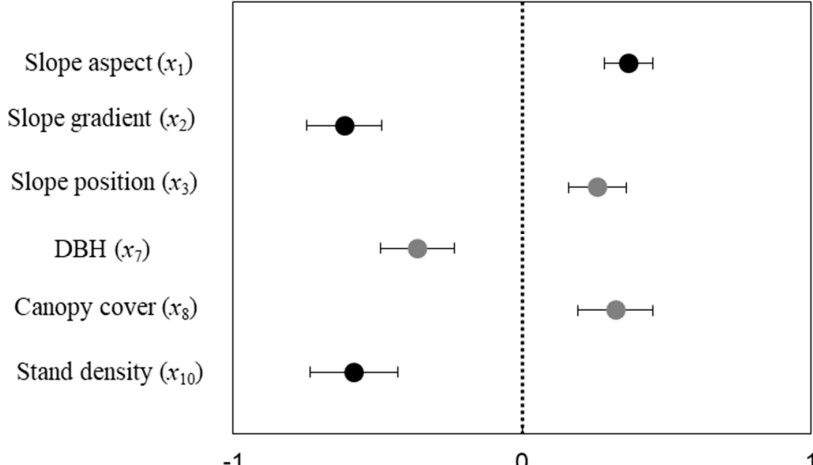

**Figure 4.** Standardized parameter estimates (±SE) of the effects of the key topography and vegetation variables on the population of *D. superans*. Black shapes indicate extremely significant effects ($p < 0.001$), grey shapes indicate significant effects ($p < 0.05$), and white shapes indicate no significance.

The $R^2$ values for the models with all variables and the key variables were 0.651 and 0.634, respectively. Additionally, the respective RMSE and MAE values were 1.043 and 1.058, and 0.828 and 0.839. The AIC values were 391.26 and 387.99, respectively, with a difference of less than 4. Furthermore, the GLMs exhibited high significance ($p < 0.001$), which indicated the reliability of both GLMs in predicting the six key variables, supporting their use to achieve the fitting effect of all topography and forest vegetation variables (Table 3).

**Table 3.** Evaluation index of the GLMs of *D. superans*.

| GLMs | Evaluation Index | | | | |
|---|---|---|---|---|---|
| | $R^2$ | *RMSE* | *MAE* | AIC | *p* |
| Model with all variables | 0.651 | 1.043 | 0.828 | 391.26 | <0.001 |
| Model with the six key variables | 0.634 | 1.058 | 0.839 | 387.99 | <0.001 |

*3.3. The Relative Importance of the Key Variables*

The categorical regression analysis of the six topography and forest vegetation variables that were identified as significantly affecting the population of *D. superans* revealed that the predictive importance ranking for the *D. superans* population was as follows: slope aspect (0.452, $p < 0.001$) > slope position (0.213, $p < 0.001$) > slope gradient (0.163, $p < 0.001$) > stand density (0.148, $p = 0.002$). Notably, the topography factors accounted for a substantial proportion of the prediction accuracy, corresponding to 82.8%, with all three topography factors exhibiting high tolerance values above 0.7, which indicated that a significant portion of their impact on the population of *D. superans* could not be explained by other independent variables (Table 4).

**Table 4.** The relative importance of categorical regression for the density of *D. superans*.

| Item | Beta | Partial Correlation | Importance | Tolerance | *p* |
|---|---|---|---|---|---|
| Slope aspect ($x_1$) | 0.429 | 0.495 | 0.452 | 0.931 | <0.001 |
| Slope gradient ($x_2$) | −0.218 | −0.249 | 0.163 | 0.735 | <0.001 |
| Slope position ($x_3$) | 0.296 | 0.367 | 0.213 | 0.941 | <0.001 |
| DBH ($x_7$) | −0.223 | −0.225 | 0.049 | 0.567 | 0.11 |
| Canopy cover ($x_8$) | 0.194 | 0.202 | −0.026 | 0.598 | 0.35 |
| Stand density ($x_{10}$) | −0.317 | −0.274 | 0.148 | 0.425 | 0.002 |

The density of *D. superans* exhibited a negative correlation with the slope gradient; as the slope gradient increased, the population size decreased. *D. superans* was predominantly found on ridges within the study area, with the lowest density on shady slopes and the highest density on sunny slopes. Moreover, the density increased as the DBH decreased, the canopy cover increased, and the stand density was reduced (Table 4 and Figure 4). As regards the slope aspect, the density of *D. superans* was minimal on shady slopes and maximal on sunny slopes. Additionally, there was a fast increase in the number of *D. superans* from shady slopes to semi-shady slopes. With respect to the slope gradient, the rate of increase in density was the highest between 6 and 15° and at values <6°, while the numbers of larvae between 6 and 15°, between 15 and 25°, and >25° were relatively similar, suggesting potential for a combined analysis. Regarding the slope location, the ridge exhibited the highest density of *D. superans* compared to the other three sites, which had comparable scores, thus indicating potential for a combined analysis (Figure 5).

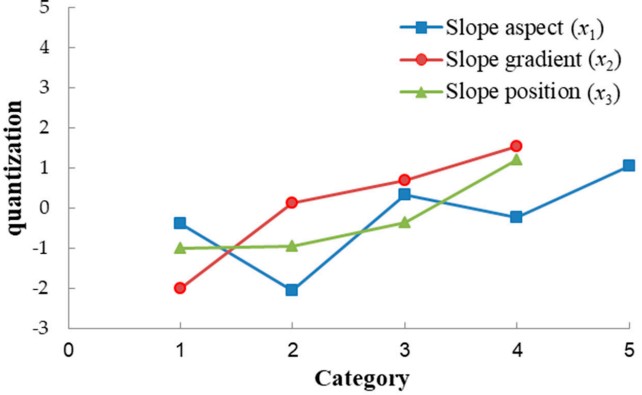

**Figure 5.** Quantification of the topography variables that were shown to significantly influence the density of *D. superans*. Category (see Table 1).

## 4. Discussion

The occurrence of *D. superans* in the southern part of the Great Xing' an Mountains appeared to be primarily influenced by the slope aspect and the herb coverage (Table 2). The probability of *D. superans* occurrence gradually increased with increasing light exposure and decreasing water availability from shady slopes to sunny slopes. A temperature increase primarily benefits egg development and larval feeding intensity, while a rise in humidity disrupts the water balance in *D. superans*, leading to increased reproduction of microorganisms such as *Beauveria bassiana* and subsequent insect mortality [9,23–25]. Studies conducted on *D. sibiricus* and *Dendrolimus pini* also revealed that an arid habitat resulting from elevated temperatures and reduced summer rainfall significantly curtailed the life cycle of these two species, thereby serving as a primary driver for their extensive outbreaks [1,26,27]. The findings of this study suggest that the outbreak risk of *D. superans* in the designated area is expected to escalate in response to climate warming. Moreover, the influence of herb coverage on the occurrence probability of this pest is likely mediated by an increase in soil respiration rate and temperature, as well as by enhanced soil fertility, which creates a more favorable environment for *D. superans* spawning. Additionally, the larvae are able to survive within the litter layer under the forest shrub grass cover, which provides protection against predation from both natural enemies and artificial control measures. The AUC value of the occurrence probability model in this study was 0.826 (Figure 2), similar to that obtained with a model for *D. superans* in the southern Great Xing' an Mountains [13]. However, in comparison to the well-established global distribution prediction model for *Cydia pomonella* (AUC = 0.94) [28], the prediction accuracy was marginally lower in this study, which is potentially attributed to the limited sample size and spatial scale and to the reduced number of variables to avoid multicollinearity. Incorporating additional dimensions such as climate and natural enemies may further enhance the model's accuracy.

In addition to the slope aspect, factors such as slope gradient, slope position, DBH, canopy cover, and stand density exerted significant influence on the population dynamics of *D. superans* (Figures 3 and 4). In particular, *D. superans* was found to occur in lower densities on steeper slopes, which aligns with the findings obtained for *Dendrolimus punctatus*. When the slope becomes more pronounced, the microclimate and microenvironment within a forest become complex, making it challenging for insect pests to proliferate due to environmental factors and natural enemies [29]. From valley to ridge, the soil water content decreases; hence, *D. superans* is better adapted to thrive on arid ridges [30]. It is worth noting that the abundance of pests does not necessarily correlate with the extent of the damage inflicted. Research conducted on *D. sibiricus* at higher latitudes revealed that forest devastation, primarily attributed to insect infestation resulting in tree mortality, predominantly occurred within a gentle slope region ranging from 11 to 13° [11,31]. Canopy cover had a significant positive impact on the population density of *D. superans*, consistent with a previous study indicating that a dense canopy intensified pest damage at the landscape level [29]. We found that a dense canopy could be beneficial for the development of *D. superans*, possibly providing attractive conditions for the larvae. Additionally, a dense crown canopy provides protection to larvae against potentially adverse weather conditions during the summer months [32]. Therefore, reducing the canopy cover by tree pruning (the trimming of branches on trees) could reduce the risk of *D. superans* infestation. The population of *D. superans* exhibited a significant decline in the high-stand-density forest, while no significant difference was found between pure and mixed larch forests. In this scenario, the resource dilution effect became more pronounced [17], possibly attributed to a reduced wind speed within the thick forest that hindered the long-distance diffusion and directional transfer of *D. superans*, particularly its larvae, predominantly in the leeward region, potentially due to the impaired flight capability of adult female moths because of the egg load in their abdomen. Adult female moths exhibit a limited flight range within leeward areas and preferentially reproduce in this specific environment [33]. The present study revealed an inverse relationship between the DBH and the population density of *D. superans*, indicating a higher prevalence of these pests in poorly growing *L. gmelinii*

forests. This finding contradicts previous hypotheses on plant vigor but can be attributed to the metabolic capacity of the main host plant, *L. gmelini*, which effectively inhibits pest growth through tannin metabolism [15,34]. Notably, our results demonstrated that tree resistance to pests correlated with the DBH, emphasizing the need for increased attention towards stunted growth and sparsely populated forests.

The results of CATREG implied that the importance of the six independent variables in predicting the population density of *D. superans* exhibited variations. The variables of slope aspect, slope gradient, slope position, and stand density played crucial roles in predicting the damage caused by *D. superans* (Table 4), with the prediction accuracy of the slope aspect reaching an impressive 45.2%. This was primarily attributed to the conducive effect of temperature on larval development, as higher temperatures resulted in reduced energy consumption for larval growth, thereby enhancing *D. superans* reproductive capacity [5]. The topography variables had a stronger effect on pest infestation than the forest vegetation ones. Generally, when the temperature drops below the threshold of 5 °C, the foraging activity of *D. superans* in its overwintering generation is significantly reduced [35]. Given that the winter temperatures in our study area fall below this threshold, there is a diminished resource dilution effect during this period. Consequently, the topography factors emerged as the primary driving force behind infection. Moreover, the tolerance of the topography variables exceeded 0.7, whereas the tolerance of the forest vegetation variables was merely 0.6 (Table 4), providing further evidence that topography not only directly influenced the population of *D. superans* but also indirectly affected it through its impact on forest vegetation.

In terms of prediction and prevention, the GLMs established in this study exhibited high accuracy and alleviated the challenges associated with data collection, thereby showing enhanced prediction and prevention capabilities. The findings from the established CATREG analysis suggested that it would be convenient to aggregate the slope grades of 6~15°, 15~25°, and >25° into a comprehensive range >6°, while conducting a separate investigation on the ridge in relation to the other three slope position variables (Figure 5). The omission of considering the influence of soil nutrients on the population dynamics of *D. superans* in our model may have contributed to its limited accuracy in predicting hazard levels. Incorporating soil factors into the model has the potential to enhance its predictive precision. Although there are still significant uncertainties regarding the impacts of climate warming on the temporal and spatial distribution of *D. superans*, its life history, and its primary hosts [1,26], current prevention and control strategies should prioritize areas with sunny slopes at low elevations during periods of drought. Additionally, reducing the understory herb coverage through increased mixed irrigation can effectively mitigate the risk of pest outbreaks. It was also reported that the population density of insects showed a significant positive correlation with the extension and shape of forest patches, indicating that large contiguous pine forests were conducive to the occurrence of insect pests [36]. This suggests that *D. superans* occurrence is a complex non-linear ecological process characterized by substantial uncertainties [37]. Further evidence is required to ascertain the factors influencing *D. superans* outbreak dynamics.

## 5. Conclusions

This study revealed differences in the topography and forest vegetation between forests of *L. gmelinii* and those infested with *D. superans*. The results may be useful for forest management in areas where *D. superans* is a major pest and where site and stand conditions are similar to those described, with topography consisting of sunny slopes, flat slopes, and ridges, and forests being sparse with limited vegetation growth. The occurrence of *D. superans* was influenced by two key factors among the 10 considered in this study, namely, slope aspect and herb coverage. Among the 10 factors considered, slope aspect, slope gradient, slope position, DBH, canopy cover, and stand density were the most significant in affecting the density of *D. superans*. Therefore, forest vegetation factors should be considered for controlling the population dynamics of *D. superans* in the future; for

example, this could be achieved by adjusting the stand density. Furthermore, to mitigate the outbreaks of *D. superans*, preventive and control measures should be prioritized in forest areas located on sunny slopes, flat slopes, and ridges. Forest management practices such as judicious thinning and regular weeding should be implemented to effectively maintain the pest population at a relatively low level.

**Author Contributions:** Conceptualization, D.H.; methodology, D.H. and J.Z.; validation, D.H. and J.Z.; formal analysis, D.H., S.W. and J.Z.; investigation, D.H., R.C. and Q.W.; resources, Q.W. All authors have read and agreed to the published version of the manuscript.

**Funding:** This research was funded by Central Research Institutes of Basic Research and Public Service Special Operations, grant number CAFYBB2020MA004.

**Data Availability Statement:** Data is contained within the article.

**Conflicts of Interest:** The authors declare no conflict of interest.

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
