# Peer review of "Evaluating Dendrolimus superans (Lepidoptera: Lasiocampidae) Occurrence and Density Modeling with Habitat Conditions"

_forests, doi:10.3390/f15020388_

Round 1
Reviewer 1 Report
Comments and Suggestions for Authors
Han et al. (XXX) investigate the influence of habitat conditions on the occurrence probability and quantity of overwintering Dendrolimus superans, aiming to provide scientific insights for effective prevention and control measures against this pest infestation. The study emphasizes the significance of habitat conditions on the occurrence probability of D. superans. The study’s well-structured design and clear language style facilitate the comprehension of its findings. In conclusion, I believe that the article would significantly contribute to the field. However, I have a few minor points that I would like to address, and I expect the authors to incorporate them in the final version. Addressing these points will enhance the article’s clarity and presentation for the readers.

Minor editing of English language required
Author Response
Response to the Comments
Dear editors and reviewers,
Re: Manuscript ID: forests-2863941 and Evaluating Larch Caterpillar (Dendrolimus superans)
Occurrence and Quantity Modeling with Habitat Conditions
Thank you for your letter and the reviewers’ comments concerning our manuscript entitled “Evaluating Larch Caterpillar (Dendrolimus superans) Occurrence and Quantity Modeling with Habitat Conditions”. Those comments are valuable and very helpful. We have read through comments carefully and have made corrections. Based on the instructions provided in your letter, we uploaded the file of the revised manuscript. Revisions in the text are shown using red highlight for additions. The corrections in the paper and our responses to the reviewer's comments are presented following.
Reviewers' Comments to Author:
Reviewer: 1
- Title: Would the authors consider enhancing the title by replacing ‘quantity’ with ‘density’? I believe this adjustment might improve the overall clarity and precision of the title. Your thoughts and decision on this suggestion are welcomed. Feel free to accept or reject accordingly.
Response: Thanks for your comment! The title of the article has been revised, and we have also employed the term "density" instead of "quantity" in the manuscript.
- Scientific Name Note: Following the initial introduction of Dendrolimus superans in the text, authors are encouraged to use the abbreviated form “D. superans” in subsequent references rather than the full genus name.
Response: Thanks for your comment! We have revised the article such that, apart from the initial occurrence of Dendrolimus superans, all other abbreviations refer to “D. superans”.
- Line 48: elevation of Picea abies Mast forest and the infestation levels of Cephalcia spp. [8]. Not sure what is it?
Response: Thanks for your comment! We have revised this sentence in the article.
- Lines 65-68: If the authors reference other studies, it is recommended to provide multiple references (at least more than one) to support the statement.
Response: We are grateful for the comment. We have added two references to the article and adjusted the order of the references.
Rhainds, M.; English-Loeb, G. Testing the resource concentration hypothesis with tarnished plant bug on strawberry: density of hosts and patch size influence the interaction between abundance of nymphs and incidence of damage. Ecological Entomology, 2003, 28(3), 348-358.
Yamamura, K. Biodiversity and stability of herbivore populations: influences of the spatial sparseness of food plants. Population Ecology, 2002, 44(1), 33- 40.
- Line 139-140: I am uncertain about using “amount” in conjunction with “occurrence” for insect
population. Kindly reconsider the phrase “occurrence amount.” Also, please review Line 169 for
consistency.
Response: Thanks for your comment! We have revised these sentences in the article.
Subsequently, the average value was computed as an indicator for the density of D. superans within each tree in each plot.
In 126 sample plots where D. superans occured, we employed the generalize linear models (GLMs) to consider the density of D. superans in sample plots as the dependent variable, while incorporating topography and forest vegetation factors as independent variables into the quantitative model.
- Please check caption of Figure 4.
Response: We are grateful for the comment. The caption of Figure 4 has been revised.
Figure 4. Quantification of topography variables that significantly influence density of D. superans.

Reviewer 2 Report
Comments and Suggestions for Authors
The paper is interesting, but there are some comments. Since the authors considered a binary logistic regression model for the description, the binary dependent variable had two values characterizing the presence and absence of insects on the sample. However, the accuracy of the measurements is unclear. Populations at low densities are characterized by strong fluctuations, and the absence of caterpillars on trees at the time of counting does not indicate whether the species can occur in a given area. In our experience, at low population densities, it is often very difficult to identify caterpillars in an area, and areas without individuals at the time of measurements according to the proposed methodology will be classified as areas unfavorable for the existence of the species, although they may be present, but at a low density. It is possible to reduce the measurement error through repeated counts, but it is unclear whether they were carried out. I would like to describe the accounting in more detail and evaluate its error. Note that the proposed procedure makes it possible to assess the suitability of an area for colonization by the pest, but does not directly allow one to assess the risks of outbreaks of the species in a certain sample plot. In fact, the occurrence of the species in a stable state is studied and the risks of outbreaks in the considered habitats are not assessed. It might be worth discussing in the discussion the problem of reliability of site classification at low insect densities.
Author Response
Response to the Comments
Dear editors and reviewers,
Re: Manuscript ID: forests-2863941 and Evaluating Larch Caterpillar (Dendrolimus superans)
Occurrence and Quantity Modeling with Habitat Conditions
Thank you for your letter and the reviewers’ comments concerning our manuscript entitled “Evaluating Larch Caterpillar (Dendrolimus superans) Occurrence and Quantity Modeling with Habitat Conditions”. Those comments are valuable and very helpful. We have read through comments carefully and have made corrections. Based on the instructions provided in your letter, we uploaded the file of the revised manuscript. Revisions in the text are shown using red highlight for additions. The corrections in the paper and our responses to the reviewer's comments are presented following.
Reviewers' Comments to Author:
Reviewer: 2
- The paperis interesting, but there are some comments. Since the authors considered a binary logistic regression model for the description, the binary dependent variable had two values characterizing the presence and absence of insects on the sample. However, the accuracy of the measurements is unclear. Populations at low densities are characterized by strong fluctuations, and the absence of caterpillars on trees at the time of counting does not indicate whether the species can occur in a given area. In our experience, at low population densities, it is often very difficult to identify caterpillars in an area, and areas without individuals at the time of measurements according to the proposed methodology will be classified as areas unfavorable for the existence of the species, although they may be present, but at a low density. It is possible to reduce the measurement error through repeated counts, but it is unclear whether they were carried out. I would like to describe the accounting in more detail and evaluate its error.Note that the proposed procedure makes it possible to assess the suitability of an area for colonization by the pest, but does not directly allow one to assess the risks of outbreaks of the species in a certain sample plot. In fact, the occurrence of the species in a stable state is studied and the risks of outbreaks in the considered habitats are not assessed. It might be worth discussing in the discussion the problem of reliability of site classification at low insect densities.
Response: Thanks for your comment! The 142 random plots, each measuring 20 m×20 m, were established in 2019, while the survey of Dendrolimus superans was conducted in 2020, thereby adhering to the statistical principle underlying the binary logistic model.
We acknowledge that low-density populations exhibit significant fluctuations. However, the primary focus of this survey is on overwintering larvae of D. superans, which possess limited mobility. Absence of D. superans presence in 16 plots suggests that insects select suitable habitats based on their ecological preferences, leading to localized and fragmented pest outbreaks within the same community. The dynamic fluctuations in populations of D. superans pose challenges for implementing repeated surveys, while the inclusion of 142 randomly selected plots effectively captures the regional situation. To enhance methodological clarity, we have incorporated a schematic diagram into our survey approach. The larval abundance of D. superans was investigated in accordance with LY/T 3030-2018, the Technical Regulation for Monitoring and Forecasting Pine Caterpillars.
In the results and discussions, we demonstrate that slope aspect exerts the most significant influence on the occurrence and density of D. superans, thereby posing a potential outbreak risk on sunny slope. Therefore, future research should prioritize emphasizing this variable.

Reviewer 3 Report
Comments and Suggestions for Authors
This research is very interesting, discussing how environmental conditions influence the presence and population of pests. The manuscript has been written well, it's just that the author needs to improve the way he writes scientific names and rewrite the introduction. I have included some of my input in the attached pdf

Author Response
Response to the Comments
Dear editors and reviewers,
Re: Manuscript ID: forests-2863941 and Evaluating Larch Caterpillar (Dendrolimus superans)
Occurrence and Quantity Modeling with Habitat Conditions
Thank you for your letter and the reviewers’ comments concerning our manuscript entitled “Evaluating Larch Caterpillar (Dendrolimus superans) Occurrence and Quantity Modeling with Habitat Conditions”. Those comments are valuable and very helpful. We have read through comments carefully and have made corrections. Based on the instructions provided in your letter, we uploaded the file of the revised manuscript. Revisions in the text are shown using red highlight for additions. The corrections in the paper and our responses to the reviewer's comments are presented following.
Reviewer: 3
This research is very interesting, discussing how environmental conditions influence the presence and population of pests. The manuscript has been written well, it's just that the author needs to improve the way he writes scientific names and rewrite the introduction. I have included some of my input in the attached pdf.
Response: Thank you for these constructive comments! Please see our response below and our revised manuscript (important revisions have been marked red).
Reviewers' Comments to Author:
Reviewer: 3
- Delete parentheses and add order and family of the insect Dendrolimus superans (Lepidoptera: Lasiocampidae)
Response: We are grateful for the comment. We revised this title “Evaluating Dendrolimus superans (Lepidoptera: Lasiocampidae) Occurrence and Density Modeling with Habitat Conditions”.
- Add a short of introduction in abstract
Response: Thanks for your comment! We have added a short of introduction in abstract.
Dendrolimus superans, a prominent forest pest in northeast China, exerts detrimental effects on tree growth and development, disrupts the ecological functioning of forests, and even alters the trajectory of succession.
- The scientific name is written in full at the beginning of the sentence. Then, the scientific name can be written as D. superans. Please check and revise throughout the manuscript.
Response: Thanks for your comment! We have revised the article such that, apart from the initial occurrence of Dendrolimus superans, all other abbreviations refer to “D. superans”.
- The introduction is not organized properly to help me understand the background and prepare me to understand the importance of the problem and objective of this study.
Basic Organization of Introductions:
- Setting - paragraph 5 can be moved as the first
- Review of Pertinent Literature - give examples of evidence of how habitat conditions affect the occurrence of a pest from related kinds of literature
- Need Statement - what’s the research question?
- Objective
and please check again the way you write the scientific name
Response: We are grateful for the comment.
- We have revised it according to the comment.
- We have added two references to the article and adjusted the order of the references.
Rhainds, M.; English-Loeb, G. Testing the resource concentration hypothesis with tarnished plant bug on strawberry: density of hosts and patch size influence the interaction between abundance of nymphs and incidence of damage. Ecological Entomology, 2003, 28(3), 348-358.
Yamamura, K. Biodiversity and stability of herbivore populations: influences of the spatial sparseness of food plants. Population Ecology, 2002, 44(1), 33- 40.
- We have added the objectives of this study were: (1) to identify the influence of topography and forest vegetative factors on the occurrence of D. superans; (2) to evaluate the relative importance of these variables and determine the key variables affecting the population dynamics of D. superans.
- Thus, the objectives of this study were: (1) to identify the influence of topography and forest vegetative variables on the occurrence of D. superans; (2) to evaluate the relative importance of these variables and determine the key variables affecting the population dynamics of D. superans, thereby providing scientific insights for effective prevention and control measures against this pest infestation.
- hm2
Response: Thanks for your comment! We have changed “hm2” to “ha”.
- Summarize the description because this information has already available in Table 1 and it’s easier to see the table.
Response: We are grateful for the comment. We have summarized the passage and presented the details in Table 1.
- Is there any figure explaining the investigation method? I think it will be better to provide an illutration for this
Response: We are grateful for the comment. We have added the diagram of investigation method.
- It is better to explain by flat, shady slope, etc. instead of using numbers in table 2
Response: We are grateful for the comment. We have revised it according to the comment.
- It is better to explain by flat, shady slope, etc. instead of using numbers in fugure 4
Response: Thanks for your comment! The values on the X-axis correspond to the categories presented in Table 1. For instance, the flat represents the slope aspect (x1) corresponding to a value of 1 on the X-axis, the valley indicates slope position(x3) corresponding to a value of 1 on the X-axis.
- Please pay attention and correct the way of writing scientific names
Response: We are grateful for the comment. We have revised them according to the comment.
